# Prevalence and Associated Factors of Musculoskeletal Disorders among Older Patients Treated at Walailak University Physical Therapy Clinic in Thailand: A Retrospective Study

**DOI:** 10.3390/ijerph21091253

**Published:** 2024-09-21

**Authors:** Chadapa Rungruangbaiyok, Parinya Vongvaivanichakul, Charupa Lektip, Wanwisa Sutara, Pathanin Jumpathong, Eiji Miyake, Keiichiro Aoki, Weeranan Yaemrattanakul

**Affiliations:** 1Department of Physical Therapy, School of Allied Health Sciences, Movement Science and Exercise Research Center, Walailak University, Nakhonsithammarat 80160, Thailand; chadapa.bn@wu.ac.th (C.R.); parinya.vo@wu.ac.th (P.V.); charupa.le@wu.ac.th (C.L.); 2Walailak Physical Therapy Clinic, School of Allied Health Sciences, Movement Science, Walailak University, Nakhonsithammarat 80160, Thailand; wanwisa.st@wu.ac.th (W.S.); pathanin.jm@wu.ac.th (P.J.); 3Department of Rehabilitation, School of Nursing and Rehabilitation Sciences, Showa University, Yokohama-shi 226-8555, Kanagawa, Japan; e.miyake@nr.showa-u.ac.jp (E.M.); k.a-0525@cmed.showa-u.ac.jp (K.A.); 4Department of Physical Therapy, Faculty of Medicine, Prince of Songkla University, Songkhla 90110, Thailand

**Keywords:** musculoskeletal disorders, older adults, prevalence, quality of life, health, Thailand

## Abstract

The prevalence of musculoskeletal disorders (MSDs) is high among older adults worldwide, significantly affecting their quality of life and overall health. Understanding the prevalence of MSDs and their associated factors is crucial to developing effective preventive and management strategies in Thailand. In this study, we aimed to investigate the prevalence of MSDs and their associated factors among older patients at Walailak University Physical Therapy Clinic. In this retrospective study, we analyzed the medical records of 396 older patients. Data on demographics, underlying diseases, career types, and treatments were collected and analyzed using descriptive statistics chi-squared tests, and logistic regression analysis to determine their associations with MSD prevalence. The overall prevalence of MSDs was 89.90%. MSD prevalence was higher among female patients than among male patients (*p* < 0.001). The most commonly affected body regions were the lower back, shoulders, and knees. Career type (*p* < 0.001) had the highest impact on the presence of MSDs after controlling for sex, age, and underlying diseases as covariates in a logistic regression model. Manual labor and heavy industry workers as well as pensioners showed an increased risk of MSDs. While older age was associated with a higher MSD prevalence using chi-squared statistics, it was removed from the logistic regression models. Pensioners were the most likely to receive treatment, indicating the need for targeted interventions for individuals with physically demanding occupations. These findings underscore the importance of targeted interventions and further research on socioeconomic factors, lifestyle behaviors, and comorbidities to manage MSDs among older adults in Thailand.

## 1. Introduction

Musculoskeletal disorders (MSDs) are a major concern among older adults worldwide, significantly affecting their quality of life and overall health [1,2,3,4,5,6]. MSDs are highly prevalent among this age group, with studies indicating that over 50% of adults aged 65 and older suffer from some form of MSD [5,6]. MSDs include various conditions, including osteoarthritis, rheumatoid arthritis, and various forms of musculoskeletal pain, which collectively contribute to reduced mobility, increased disability, and a greater dependency on healthcare services [7,8,9,10,11,12,13,14]. MSDs are the leading cause of chronic pain in older adults, with studies indicating that these conditions are responsible for nearly 50% of all cases of chronic pain in this age group. This chronic pain is associated with significant decreases in mobility and quality of life [15]. Moreover, MSDs have a profound effect on the mobility and overall well-being of older people. According to Stubbs, Schofield, and Patchay (2016), there is a significant association between MSD-related pain and mobility limitations in older adults [16]. It can lead to increased dependency on others, reduced ability to perform daily activities, and a greater risk of social isolation. These limitations not only diminish quality of life but also exacerbate the risk of further health complications, including falls and their associated morbidity and mortality [17]. MSDs in older adults are primarily influenced by aging, obesity, gender differences, physical inactivity, genetic predisposition, and socioeconomic factors [18,19,20,21,22]. With the global population aging, the prevalence of MSDs among older adults is expected to increase. The proportion of older adults (aged ≥ 65 years) in the global population is increasing rapidly because of the decline in fertility rates and improvements in healthcare. According to the United Nations, the number of individuals aged ≥ 65 years is expected to double by 2050, with the fastest growth projected in developing countries [23]. Because of the heterogeneity of MSDs, a detailed understanding of the local prevalence and associated factors is crucial to developing effective preventive and management strategies in local healthcare systems.

MSDs have a particularly high prevalence observed in populations with a history of physically demanding occupations. Numerous studies have documented the prevalence and risk factors of work-related musculoskeletal disorders in healthcare professionals, highlighting the impact of repetitive movements, awkward postures, and heavy lifting on the development of these conditions [24,25]. However, less is known about the prevalence of MSDs in retired populations, who may continue to experience the effects of occupational strain. This study aimed to fill this gap by analyzing the prevalence and associated factors of MSDs among older adults attending a physical therapy clinic. 

In Thailand, the demographic shift towards an aging population has increased the focus on geriatric health issues, including MSDs. The prevalence of MSDs among older adults in Thailand is a critical area of research, particularly in clinical settings, where targeted interventions can be developed and implemented [26,27,28]. Therefore, in this retrospective study, we aimed to analyze the prevalence of MSDs and their associated factors among older patients treated at the Walailak University Physical Therapy Clinic, a major outpatient healthcare center in Thailand. This analysis can provide valuable insights for clinicians, policymakers, and researchers into developing targeted interventions and healthcare strategies.

By examining patient records and demographic data, we aim to contribute to the body of knowledge regarding the prevalence of MSDs among older adults in Thailand, ultimately providing evidence-based practices and improving the delivery of healthcare services to this vulnerable population.

## 2. Materials and Methods

### 2.1. Study Population

In this retrospective study, we analyzed the medical records of older patients (aged ≥ 60 years) who visited the Walailak University Physical Therapy Clinic in Thailand between January 2018 and December 2022. The clinic serves a diverse population, primarily from the Nakhon Si Thammarat province, which is the southern part of Thailand, with a range of musculoskeletal conditions and neurological conditions.

### 2.2. Inclusion and Exclusion Criteria

Only medical records of patients aged ≥ 60 years were included in this study, whereas those of patients with incomplete data, immune-related diseases, or a diagnosis of cancer were excluded. Additionally, patients with immune-related diseases or a diagnosis of cancer were excluded to minimize confounding factors, as these conditions could independently influence the musculoskeletal health outcomes and therapy needs. The inclusion and exclusion criteria were strictly adhered to during the data screening process, which was conducted by three physiotherapists.

### 2.3. Data Collection

Data were collected from patient records (in tangible copies and from the database of the physical therapy clinic) by three physiotherapists at Walailak University Physical Therapy Clinic. The data included patients’ baseline information, such as age, sex, occupational status, and address information, and medical data, including diagnosis of a disease, medical treatment, physical therapy received, number of physical therapy visits, and frequency of physical therapy visits.

### 2.4. Statistical Analysis

Exploratory analyses were performed using the chi-squared or Fisher’s exact test for categorical variables. Variables that were found to be associated with the presence of MSDs were subsequently included in a logistic regression analysis using a binary logistic response model to identify possible interactions and to estimate effect sizes. Statistical significance was set at *p* = 0.05 (two-sided), and statistical analyses were performed using IBM SPSS 25.0.0.0 (IBM, Armonk, NY, USA) unless otherwise specified.

### 2.5. Ethical Consideration

This study was approved by the Ethics Committee of Walailak University and followed the principles of the Declaration of Helsinki (approval number: WUEC-23-113-01).

## 3. Results

### 3.1. Demographics and Characteristics of Participants

Patient demographics and characteristics are presented in Table 1. In total, 396 patients were included in this study, and their medical records were analyzed. Of these patients, 146 (36.87%) were male and 250 (63.13%) were female. Regarding the age distribution, 322 (81.31%), 58 (14.65%), and 16 (4.04%) patients were in the young-old (60–74 years), middle-old (75–84 years), and oldest-old (≥85 years) categories, respectively. Regarding underlying diseases, 233 (58.84%) patients had at least one underlying disease, with hypertension (38.89%), dyslipidemia (27.78%), and type 2 diabetes mellitus (15.15%) being the most common conditions. The distribution of career types showed that pensioners represented the largest group at 36.11%, followed by manual laborers (18.18%) and business owners (16.67%).

### 3.2. Prevalence of Musculoskeletal Disorders and Its Association with Demographical Factors

The overall prevalence of MSDs in the study population was 89.90% (n = 356/396), and the prevalence of MSDs was associated with sex (*p* < 0.001), age (*p* = 0.013), underlying disease (*p* = 0.028), and career type (*p* < 0.001) by a chi-squared test analysis (Table 2). Binary logistic regression analyses including patient sex, patient age, the presence of any underlying disease, and career type as categorical predictors showed that patient age had a relevant overlap with the other three cofactors and thus was removed from the model. Patient sex, career type, and the presence of any underlying disease, however, all had a significant independent effect on the prevalence of MSDs. The risk of developing MSDs was 5.2-fold higher in women than in men after controlling for the other covariates in the model. With respect to the different career types, manual labor and heavy industry workers as well as pensioners were at an increased risk of developing MSDs. The magnitude of these conditions was much higher than any other factor, with a 19.5-fold increase in the prevalence of MSDs in pensioners and a 7.9-fold increase in manual labor and heavy industry workers compared with unemployed patients. In contrast to these effect sizes, the impact of the presence of underlying diseases on the prevalence of MSDs was relatively small and the correlation was inverse, i.e., the presence of an underlying disease was associated with a lower prevalence of MSDs. A more detailed analysis on the impact of underlying diseases, including the three most common underlying conditions (hypertension, dyslipidemia, and type 2 diabetes), as separate cofactors in the linear regression model showed that only hypertension had an effect. Patients with hypertension had a slightly lower prevalence of MSDs (0.39-fold decrease, *p* = 0.024;) (Table 3).

Table 4 presents a detailed classification based on body region and sex. The lower back was the most frequently affected area (22.98%), followed by the shoulders (16.16%) and knees (13.13%). The prevalence of MSDs was higher among female patients (59.60%) than among male patients (30.56%).

Regarding age-based classification (Table 5), the prevalence of MSDs was the highest among patients in the young-old category (60–74 years; 75%), followed by the middle-old (75–84 years; 12.12%) and oldest-old (≥85 years; 3.03%) categories.

Table 6 presents the prevalence of MSDs based on body region and the presence of underlying diseases. The prevalence of MSDs was higher among patients with underlying diseases (58.84%) than among patients without underlying diseases (41.16%). The lower back, shoulders, and knees were the most affected areas in both groups. Table 7 presents the presence of underlying diseases of patients with musculoskeletal disorders. Hypertension was the most common underlying disease, present in 36.80% of patients with MSD, followed by dyslipidemia (27.25%) and type 2 diabetes (13.76%).

Table 8 presents the prevalence of MSDs based on body region and career type. Notably, the prevalence of MSDs was highest among pensioners (36.11%) and manual workers (18.18%). The lower back, shoulders, and knees were the most commonly affected areas across all career types.

### 3.3. Healthcare Utilization Patterns

The most common diagnoses among patients with MSDs were muscle pain/strain (27.53%), degenerative conditions, such as spondylosis and spinal stenosis (20.51%), and lower back pain (12.64%). The treatment modalities included superficial heat therapy (94.38%), ultrasound therapy (83.71%), and massage (57.30%). The frequency of visits to the clinic varied, with most patients attending the clinic twice monthly (38.48%) (Table 9).

## 4. Discussion

### 4.1. Prevalence of MSDs 

The prevalence of MSDs among older patients at Walailak University Physical Therapy Clinic was high (89.90%), which is consistent with previously published data documenting the widespread nature of MSDs among older adults. This high prevalence underscores the significant burden that MSDs pose on older adults, affecting their mobility, quality of life, and overall health. 

The data from Table 4 to Table 8 further illustrate the specific patterns of MSD prevalence across different body regions, demographic groups, and occupational backgrounds. For instance, Table 4 shows that lower back pain was the most prevalent condition, affecting 34.09% of patients, which aligns with global estimates. In their systematic review, de Souza et al. (2021) [1] reported that the prevalence of lower back pain among older adults ranged from 21% to 75%, depending on the population studied and the methods used for assessment. This variability in prevalence underscores the importance of localized studies like ours to understand the specific burden of MSDs in different contexts.

Moreover, our study revealed significant differences in MSD prevalence by sex, age, and career type. As indicated in Table 4, women were more affected by MSDs than men (94.00% vs. 83.56%), which is consistent with findings from global studies that women are more susceptible to certain MSDs, owing to factors such as hormonal changes and bone density loss [29,30]. Additionally, Table 5 highlights that older age groups, particularly those aged 75–84 years, exhibited higher prevalence rates, reflecting the cumulative impact of aging on musculoskeletal health [31].

Occupational history also played a crucial role, as evidenced by the data in Table 8. Pensioners and manual laborers were among the most affected groups, with MSD prevalence rates of 94.37% and 95.83%, respectively. This is consistent with studies showing that physically demanding jobs contribute to the long-term risk of MSDs [32]. These findings emphasize the need for targeted interventions aimed at preventing and managing MSDs, particularly among high-risk groups identified by career type and age.

The significant effect of MSDs on disability and quality of life in the elderly population has been well-documented. March et al. (2014) [2] investigated the global burden of disability due to MSDs and highlighted the profound effect these conditions have on older adults. Our findings align with this global perspective, demonstrating that MSDs are a leading cause of disability and reduced quality of life among the elderly population in Thailand. The data presented in this study reinforce the urgent need for comprehensive management strategies tailored to the specific needs of older adults, particularly in regions with rapidly aging populations.

### 4.2. Sex and Age Differences

In the present study, a significant association was observed between sex and the prevalence of MSDs, with the prevalence being higher among female patients than among male patients. This association reflects broader epidemiological patterns, with a higher prevalence of musculoskeletal pain and MSDs observed among females [33,34,35,36,37]. Women are biologically and physiologically more susceptible to certain MSDs [29,30]. Hormonal differences, particularly the effects of estrogen, play a crucial role in this disparity between both sexes. Estrogen influences joint and muscle health, and its decline during menopause can increase susceptibility to osteoarthritis and osteoporosis [38]. Occupational roles and lifestyle choices may contribute to the disparity between both sexes among individuals with MSDs. Women are more likely to engage in repetitive and physically demanding tasks in occupational and domestic settings, which can contribute to higher rates of musculoskeletal strain and injury [39,40]. 

The higher prevalence of MSDs in the young-old category (60–74 years) than in the other categories might be attributed to higher levels of physical activity, which result in increased wear and tear of musculoskeletal structures. Individuals aged 60–74 years often maintain higher levels of physical activity, including occupational and recreational activities, than those in older age groups [41,42]. However, logistic regression analysis showed that age differences were no longer relevant when sex and career type were taken into account. The career group “pensioners” might have a considerable impact on this result.

### 4.3. Underlying Diseases 

This study revealed a significant although relatively small and unexpectedly inverse association between the presence of arterial hypertension and the prevalence of musculoskeletal disorders (MSDs) among the older adults attending the Walailak University Physical Therapy Clinic. Of the patients with MSDs, 36.80% had hypertension as at least one underlying condition. Dyslipidemia (27.25%) and type 2 diabetes (13.76%) were the second and third most common comorbidities. Comorbidities can contribute to the onset, progression, and severity of MSDs through various mechanisms, including systemic inflammation, altered biomechanics, and reduced physical activity. Chronic diseases, such as diabetes, cardiovascular diseases, and autoimmune disorders, can lead to systemic inflammation, which can exacerbate the severity of MSDs [43,44,45]. The inflammatory cytokines released in cases of these conditions can contribute to joint and muscle degeneration, thereby increasing the risk and severity of MSDs [46]. However, our data suggest that in contrast to decades of manual labor, the effect of these underlying diseases on the prevalence of MSDs is relatively small. The inverse correlation moreover might indicate that healthcare utilization because of common underlying diseases might have preventive effects.

### 4.4. Impact of Socioeconomic Factors on Musculoskeletal Disorders (MSDs)

The findings of this study highlight the significant influence of socioeconomic factors, particularly occupational status and career type, on the prevalence of MSDs among older adults. Notably, the study revealed that pensioners exhibited the highest prevalence of MSDs, which may be attributed to the cumulative effects of years of physical labor and the subsequent onset of musculoskeletal issues in retirement. Although pensioners in Thailand generally experience financial stability and have more leisure time, the persistence of MSDs in this group suggests that long-term occupational strain continues to affect their musculoskeletal health. Similar findings have been reported in other studies, where the lingering effects of physically demanding careers have been linked to a higher incidence of MSDs in retirees [31,32].

Moreover, this study found significant associations between career type and the prevalence of MSDs, with manual laborers showing a higher risk than other occupational groups. This correlation is consistent with the literature, which indicates that prolonged exposure to physically demanding tasks significantly increases the risk of developing musculoskeletal conditions. Manual laborers, in particular, are prone to repetitive strain injuries and cumulative trauma disorders, which can exacerbate MSDs over time [24,25]. Studies like those by Picavet and Schouten (2003) and Coggon et al. (2013) report that the 12-month prevalence of MSDs remains high in older populations, especially among those with a history of manual labor, reflecting the persistent nature of these conditions [47,48]. Similarly, studies on work-related musculoskeletal disorders among healthcare professionals, such as physical therapists, dentists, and surgeons, have demonstrated that careers involving repetitive movements, awkward postures, and heavy lifting contribute to a higher prevalence of MSDs, particularly in the lower back, neck, and shoulders [49,50].

In contrast, the proportion of patients who were self-paying for treatment was almost equal to those who were not, indicating that financial barriers might not significantly limit access to physical therapy services among the older adults in this study. This suggests that in settings where healthcare services are accessible and affordable, socioeconomic status may not be a primary determinant of healthcare utilization. However, the persistence of high MSD prevalence across various socioeconomic groups underscores the importance of addressing occupational risks and ensuring equitable access to preventive and therapeutic interventions for all older individuals.

Given these findings, it is crucial to develop targeted interventions that address the specific occupational risks associated with different career types, especially those that involve significant physical demands. Interventions should focus on promoting ergonomic practices, providing adequate support during physically demanding tasks, and offering early intervention and rehabilitation services to mitigate the long-term effects of occupational strain. Furthermore, policies aimed at improving access to healthcare for all socioeconomic groups, particularly those in lower income brackets, could help reduce the overall burden of MSDs on the older population.

### 4.5. Implications for Clinical Practice

The findings of this study have important implications for clinical practice and public health strategies in Thailand. The high prevalence of MSDs and their association with demographical factors suggest the need for tailored preventive and management strategies, including ergonomic interventions, regular physical activity programs, and targeted physiotherapy, to effectively manage and prevent MSDs.

### 4.6. Limitations and Future Research

This study had some limitations. First, the assessment of the prevalence of MSDs was conducted among older individuals attending a physiotherapy service, which inherently selected for a population already experiencing musculoskeletal issues. This selection bias likely contributed to the high prevalence of MSDs observed in the study, as those without significant musculoskeletal problems are less likely to seek physiotherapy care. Consequently, the findings may not be fully generalizable to the broader elderly population, particularly those who do not seek or require physiotherapy services. Second, the retrospective design of the study may have been subject to biases in record keeping and data collection, potentially affecting the reliability and accuracy of the data. Prospective study designs should be incorporated into future research to validate our findings and explore the effectiveness of specific interventions. Third, this study was conducted at a single physical therapy clinic, Walailak University Physical Therapy Clinic, which may limit the generalizability of our results to other populations or clinical settings in Thailand. Fourth, owing to the lack of longitudinal data, we could not assess changes in the prevalence of MSDs and the long-term effect of treatments over time. Future studies in which the impact of socioeconomic factors and lifestyle behaviors on the prevalence of MSDs are investigated are warranted to provide a more comprehensive understanding of the prevalence of MSDs among older adults in Thailand. Finally, the potential influence of comorbidities on the prevalence and severity of MSDs was not extensively analyzed in this study, indicating the need for a more detailed investigation of the interplay between various health conditions and MSDs. 

## 5. Conclusions

This study reveals a notably high prevalence of MSDs among older patients at Walailak University Physical Therapy Clinic, with an overall prevalence of 89.90%. The significant associations between MSD prevalence and factors such as sex, age, underlying diseases, and career type highlight the critical need for targeted interventions. Women, older adults, and individuals with physically demanding career histories, such as manual laborers and pensioners, are particularly at risk. These findings underscore the importance of developing evidence-based practices and public health policies that are specifically tailored to address the diverse associated factors contributing to MSDs in older adults. By implementing targeted prevention and management strategies, healthcare providers can improve the quality of life for this vulnerable population in Thailand. Furthermore, these results emphasize the need for continued research into the effect of socioeconomic factors, lifestyle behaviors, and comorbidities on MSD prevalence and severity to better inform future interventions and healthcare policies.

## Figures and Tables

**Table 1 ijerph-21-01253-t001:** Demographics and characteristics of the participants (N = 396).

Items	Male (N = 146)		Female (N = 250)		Total (N = 396)	
	n	%	n	%	n	%
Age (years)						
Young-old (60–74)	121	82.88	201	80.40	322	81.31
Middle-old (75–84)	21	14.38	37	14.80	58	14.65
Oldest-old (>85)	4	2.74	12	4.80	16	4.04
Underlying disease						
No	62	42.47	101	40.40	163	41.16
Yes	84	57.53	149	59.60	233	58.84
Hypertension	56	38.36	98	39.20	154	38.89
Dyslipidemia	30	20.55	80	32.00	110	27.78
Type 2 diabetes	26	17.81	34	13.60	60	15.15
Heart disease	7	4.79	5	2.00	12	3.03
Allergic reaction	0	0.00	4	1.60	4	1.01
Asthma	0	0.00	4	1.60	4	1.01
Thyroid disease	3	2.05	3	1.20	6	1.52
Other	8	5.48	8	3.20	16	4.04
Career						
Manual Labor and Heavy Industry	35	23.97	37	14.80	72	18.18
Office and Administrative Work	5	3.42	10	4.00	15	3.79
Pensioner	54	36.99	89	35.60	143	36.11
Business Owner	29	19.86	37	14.80	66	16.67
Housewife	0	0	30	12.00	30	7.58
Unemployed	23	15.75	47	18.80	70	17.68
Self-pay						
No	83	56.85	139	55.60	222	56.06
Yes	63	43.15	111	44.40	174	43.94

**Table 2 ijerph-21-01253-t002:** Association between patients’ demographic and the prevalence of musculoskeletal disorders (chi-squared test).

Parameters	*X* ^2^	*p* Value
Sex	12.56	<0.001 *
Age	8.62	0.013 *
Underlying disease	4.80	0.028 *
Career	33.02	<0.001 *
Self-pay	0.23	0.632

*p*-value determined using Pearson chi-squared test, * statistical significance (*p* < 0.05).

**Table 3 ijerph-21-01253-t003:** Association between patients’ demographics and the prevalence of musculoskeletal disorders (binary logistic regression model including risk factors like sex, age, underlying disease, and career type).

Parameters	Exp(B)	*p* Value
Sex (female vs. male)	5.208	<0.001 *
Age 60–74 years 75–84 years >85 years (RC)		0.932 0.709 0.780
Underlying disease (absent vs. present)	3.167	0.007 *
Career		<0.001 *
Manual labor and heavy industry Office and administrative work Pensioner Business owner Housewife Unemployed (RC)	7.292 19.476 3.523	0.001 * 0.999 <0.001 * 0.021 * 0.377

N = 396, likelihood ratio X^2^ = 56.537, *p* < 0.001. N indicates number of cases included, Exp(B) indicates effect size, RC indicates reference category and * indicates statistical significance (*p* < 0.05).

**Table 4 ijerph-21-01253-t004:** Prevalence of musculoskeletal disorders among older patients at Walailak Physical Therapy Clinic based on body region and sex.

Body Site	All Participants (n = 396)		Male (n = 146)		Female (n = 250)	
	n	(%)	n	(%)	n	(%)
Overall patients with musculoskeletal disorders	356	89.90	121	30.56	236	59.60
Neck	45	11.36	20	5.05	25	6.31
Shoulders	64	16.16	23	5.58	41	10.35
Ankles/Feet	12	3.03	4	1.01	8	2.02
Hands/Wrists	12	3.03	4	1.01	8	2.02
Upper back	30	7.58	6	1.52	24	6.06
Lower back	91	22.98	33	8.33	58	14.65
Hips/Thighs	45	11.36	15	3.79	30	7.58
Knees	52	13.13	14	3.54	38	9.60
Elbows	6	1.52	2	0.51	4	1.01

**Table 5 ijerph-21-01253-t005:** Prevalence of musculoskeletal disorders among older patients at Walailak Physical Therapy Clinic based on body region and age.

Body Site	All Participants (n = 396)		Young-Old (n = 322)		Middle-Old (n = 58)		Oldest-Old (n = 16)	
	n	(%)	n	(%)	n	(%)	n	(%)
Overall patients with musculoskeletal disorders	356	89.90	297	75.00	48	12.12	12	3.03
Neck	45	11.36	36	9.09	8	2.02	1	0.25
Shoulders	64	16.16	52	13.13	9	2.27	3	0.76
Ankles/Feet	12	3.03	10	2.53	2	0.51	0	0.00
Hands/Wrists	12	3.03	12	3.03	0	0.00	0	0.00
Upper back	30	7.58	23	5.81	7	1.77	0	0.00
Lower back	91	22.98	76	19.19	11	2.78	4	1.01
Hips/Thighs	45	11.36	39	9.85	5	1.26	1	0.25
Knees	52	13.13	43	10.86	6	1.52	3	0.76
Elbows	6	1.52	6	1.52	0	0.00	0	3.03

**Table 6 ijerph-21-01253-t006:** Prevalence of musculoskeletal disorders among older patients at Walailak Physical Therapy Clinic based on body region and the presence of underlying diseases.

Body Site	All Participants (n = 396)		Underlying Disease			
			No		Yes	
	n	(%)	n	(%)	n	(%)
Overall patients with musculoskeletal disorders	356	89.90	153	38.64	204	51.52
Neck	45	11.36	20	5.05	25	6.31
Shoulders	64	16.16	27	6.82	37	9.34
Ankles/Feet	12	3.03	6	1.52	6	1.52
Hands/Wrists	12	3.03	9	2.27	3	0.76
Upper back	30	7.58	15	3.79	15	3.79
Lower back	91	22.98	45	11.36	46	11.62
Hips/Thighs	45	11.36	15	3.79	30	7.58
Knees	52	13.13	13	3.28	39	9.85
Elbows	6	1.52	3	0.76	3	0.76

**Table 7 ijerph-21-01253-t007:** The presence of underlying diseases in patients with musculoskeletal disorders.

Underlying Disease	Total (N = 356)	
	n	%
Hypertension	131	36.80
Dyslipidemia	97	27.25
Type 2 diabetes	49	13.76
Heart disease	11	3.09
Allergic reaction	4	1.12
Asthma	4	1.12
Thyroid disease	5	1.40
Other	12	3.37

**Table 8 ijerph-21-01253-t008:** Prevalence of musculoskeletal disorders among older patients at Walailak Physical Therapy Clinic based on body region and career type.

Body Site	All Participants (n = 396)		Career											
			Manual Worker		Office and Administrative Work		Pensioner		Business Owner		Housewife		Unemployed	
	n	(%)	n	(%)	n	(%)	n	(%)	N	(%)	n	(%)	n	(%)
Overall patients with musculoskeletal disorders	356	89.90	66	16.67	15	3.79	140	35.35	58	14.65	27	6.82	51	12.88
Neck	45	11.36	11	2.78	0	0.00	15	3.79	7	1.77	5	1.26	7	1.77
Shoulders	64	16.16	9	2.27	4	1.01	24	6.06	12	3.03	4	1.01	11	2.78
Ankles/Feet	12	3.03	3	0.76	0	0.00	4	1.01	2	0.51	0	0.00	3	0.76
Hands/Wrists	12	3.03	3	0.76	1	0.25	6	1.52	1	0.25	1	0.25	0	0.00
Upper back	30	7.58	4	1.01	1	0.25	14	3.54	5	1.26	1	1.01	5	1.26
Lower back	91	22.98	13	3.28	6	1.52	40	10.10	14	3.54	4	1.52	14	3.54
Hips/Thighs	45	11.36	11	2.78	1	0.25	15	3.79	8	2.02	6	1.26	4	1.01
Knees	52	13.13	10	2.53	2	0.51	20	5.05	9	2.27	5	0.25	6	1.52
Elbows	6	1.52	2	0.51	0	0.00	2	0.51	0	0.00	1	6.82	1	0.25

**Table 9 ijerph-21-01253-t009:** Healthcare utilization patterns among older patients with MSDs.

Healthcare Service	Male (N = 128)		Female (N = 228)		Total (N = 356)	
	n	%	n	%	n	%
Diagnosis						
Muscle: MPS/Muscle strain	36	10.11	62	17.42	98	27.53
Degenerative: Spondylosis/Spondylolisthesis/Spinal stenosis	23	6.46	50	14.04	73	20.51
Low back pain	17	4.78	28	7.87	45	12.64
Osteoarthritis	20	5.62	23	6.46	43	12.08
Adhesive capsulitis	7	1.97	29	8.15	36	10.11
Tendinitis	13	3.65	10	2.81	23	6.46
Joint pain/stiffness	5	1.40	10	2.81	15	4.21
Plantar fasciitis	2	0.56	5	1.40	7	1.97
Ligament sprain	4	1.40	2	0.56	6	1.69
Trigger finger	1	0.28	4	1.12	5	1.40
Carpal tunnel syndrome	0	0.00	3	0.84	3	0.84
Other	0	0.00	2	0.56	2	0.56
Types of treatments						
Superficial Heat Therapy	123	34.55	213	59.83	336	94.38
Superficial Cold Therapy	2	0.56	8	2.25	10	2.81
Short Wave Diathermy	12	3.37	19	5.34	31	8.71
Ultrasound Therapy	104	29.21	194	54.49	298	83.71
Transcutaneous Electrical Nerve Stimulation (TENS)	28	7.87	39	10.96	67	18.82
Interferential Current Therapy (IFC)	23	6.46	24	6.74	47	13.20
Mobilization	33	9.27	52	14.61	85	23.88
Massage	70	19.66	134	37.64	204	57.30
Manual Therapy	8	2.25	12	3.37	20	5.62
Exercise Therapy	18	5.06	35	9.83	53	14.89
Passive Stretching	83	23.31	149	41.85	232	65.17
Traction	10	2.81	22	6.18	32	8.99
Home Program	2	0.56	8	2.25	10	2.81
Frequency of visits						
Once/month	43	12.08	64	17.98	107	30.06
Twice/month	52	14.61	85	23.88	137	38.48
Thrice/month	24	6.74	53	14.89	77	21.63
Four times/month	9	2.53	20	5.62	29	8.15
Five times/month	0	0.00	3	0.84	3	0.84
Six times/month	0	0.00	3	0.84	3	0.84

## Data Availability

The raw data supporting the conclusions of this study will be made available by the authors upon request.

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
