# Peer review of "Prevalence and Associated Factors of Musculoskeletal Disorders among Older Patients Treated at Walailak University Physical Therapy Clinic in Thailand: A Retrospective Study"

_ijerph, 2024, doi:10.3390/ijerph21091253_

Round 1
Reviewer 1 Report
Comments and Suggestions for Authors
I want to thank you the authors for the submission. However, I consider the paper has several weaknesses and limitations, which impact the quality and suitability of the manuscript for publication in its present form.
General Comments:
Title: I recommend the authors consider substituting "risk factors" with "associated factors" in both the title and throughout the manuscript. Given the nature of the study design, it is more accurate to refer to associations rather than risks.
Introduction: The introduction could benefit from enhancement. I suggest the authors incorporate elements that underscore the significance and innovation of the data presented. A more robust justification for conducting the study is also advisable. The assessment of the prevalence of musculoskeletal disorders among elderly individuals attending a physiotherapy service appears somewhat anticipated and might be prone to substantial bias, which could account for the high prevalence reported. Furthermore, the analysis of associated factors was somewhat limited, which restricts the depth of the conclusions.
Method: The statistical analysis employed is relatively basic, relying solely on bivariate tests. It would be valuable to include effect size values for the observed associations. I recommend considering the use of multiple logistic regression models. Additionally, the methodology section would benefit from a more detailed description. A more thorough explanation of the methods used would enhance the clarity and replicability of the study.
Reviewer 2 Report
Comments and Suggestions for Authors
Dear Authors,
Kindly address the following:
1. Data were collected from patient records (in tangible copies and from the database 72 of the physical therapy clinic) by a physiotherapist at Walailak University Physical Therapy Clinic- In regard to this statement, whether the entire data data is collected by a single Physio?
2. Women 148 are biologically and physiologically more susceptible to certain MSDs. can you provide reference for this statement.
3. The impact of socio-economic factors in this study was mentioned as it is related to MSDs. This need to be addressed.
4. The underlying diseases correlation with MSDs need to elaborated as this is one of the main point in this study.
Reviewer 3 Report
Comments and Suggestions for Authors
The aim of this research is to study the prevalence of MSDs and their associated factors in elderly patients at Walailak University Physiotherapy Clinic. The medical records of 396 elderly patients are analyzed. Demographic data, underlying diseases, career types and treatments were collected and analyzed using descriptive statistics and chi-square tests to determine their associations with the prevalence of MSDs. The main results are: The overall prevalence of MSDs was 89.90%; the prevalence of MSDs was higher in women than in men (p<0.001). The body regions most frequently affected were the lower back, shoulders and knees. MSD prevalence was significantly associated with gender (p<0.001), age (p=0.013), underlying diseases (p=0.028) and career type (p<0.001). The authors conclude that these findings underscore the importance of targeted interventions and further research into socioeconomic factors, lifestyle behaviors and comorbidities to manage MSDs in older adults in Thailand.
The restrospective study presented is of interest to the community. The article is well structured, the methodological approach clear and the discussion pertinent. The objective is well defined and the hypotheses well formulated. However, the following comments and recommendations could further improve the article:
- In the “introduction” section: Some references on the risks and prevalence of WMSD could be added. Some studies have proposed a meta-analysis of total prevalence and prevalence by body area, sex, age, presence of underlying diseases and career type for health professionals or dentists worldwide...., could show the interest and above all the originality of the work presented. In fact, the choice of information studied and the parameters likely to have a link or an effect on prevalence have already been used in the literature. It would be interesting to justify the work presented in relation to these works. For example:
1) Work-Related Musculoskeletal Disorders in Iranian Dentists: A Systematic Review and Meta-analysis., Saf Health Work. 2018 Mar;9(1):1-9.
2) P Gorce, J Jacquier-Bret, A systematic review of work related musculoskeletal disorders among physical therapists and physiotherapists, Journal of Bodywork and Movement Therapies, 2024
, Effect of Assisted Surgery on Work-Related Musculoskeletal Disorder Prevalence by Body Area among Surgeons: Systematic Review and Meta-Analysis., Int J Environ Res Public Health. 2023 Jul 20;20(14),
4) Occupational ergonomics and related musculoskeletal disorders among dentists: A systematic review. Work. 2023;74(2):469-476.
5) R. Tavakkol, E. Kavi, S. Hassanipour, H. Rabiei, M. Malakoutikhah, The global prevalence of musculoskeletal disorders among operating room personnel: a systematic review and meta-analysis, Clin Epidemiol Glob Heal [Internet] 8 (4) (2020) 1053–1061,
6) Efficacy of Interventions in Reducing the Risks of Work-Related Musculoskeletal Disorders Among Healthcare Workers: A Systematic Review and Meta-Analysis. Workplace Health Saf. 2023 Dec;71(12):557-576.
7) Work-related musculoskeletal disorders among physical therapists: A systematic review., J Back Musculoskelet Rehabil. 2016 Aug 10;29(3):417-28.
8) J Jacquier-Bret, P Gorce, Prevalence of body area work-related musculoskeletal disorders among healthcare professionals: a systematic review, International Journal of Environmental Research and Public Health 20 (1), 841
The introduction could be more complete and interesting for the reader.
- Some of these references could also be used in the discussion section to address other aspects of the study, such as high-prevalence body zones, the effect of the methods used, or the problem of data homogeneity, or the choice of parameters having an effect on MSD prevalence, ...etc.
- The discussion could be supported by a larger number of bibliographical references to propose an analysis, recommendations and relevant limitations that can be exploited by the international scientific community. For example, the results of the proposed work could be compared, discussed and commented on with recent work (systematic review and meta-analysis) in the healthcare professional community, surgeons, physiotherapists, operating room personnel or nurses etc. This could increase the relevance and quality of the discussion, and therefore the interest of the readers.
Or discuss the fact that the data in tables 3 to 7 are derived from medical data indicate the Prevalence of musculoskeletal disorders among older patients at Walailak Physical Therapy Clinic based on body region and sex, age, presence of underlying diseases and career type. I think it would be interesting to extend the discussion on how the information obtained in this study compares with that in the literature, which is often obtained using questionnaires.
It might be interesting to discuss the prevalence obtained 12 month, career or other (comparison of data from the present study with the literature).
Round 2
Reviewer 1 Report
Comments and Suggestions for Authors
After reading the response letter and the revised version of the manuscript, I believe that the authors have successfully addressed all previous comments. Therefore, I consider that the manuscript meets the minimum requirements for suitability for publication.
Reviewer 3 Report
Comments and Suggestions for Authors The authors have taken into account the comments, questions and recommendations. The article has been significantly improved. I propose to accept in current form